# Inappropriate Corporate Strategies: Latin American Companies That Increase Their Value by Short-Term Liabilities

**Jorge Feregrino [1], Juan Felipe Espinosa-Cristia [2,*] , Nelson Lay [3] and Luis Leyton [4]**

1    Fes Acatlán, UNAM, México City 53150, Mexico
2    Departamento de Ingeniería Comercial, Universidad Técnica Federico Santa María, Valparaíso 2390123, Chile
3    Facultad de Educación y Ciencias Sociales, Universidad Andres Bello, Viña del Mar 2531015, Chile
4    Facultad de Economía y Negocios, Universidad Andres Bello, Viña del Mar 2531015, Chile
*     Correspondence: juan.espinosacr@usm.cl

**Abstract:** This study seeks to understand the financing strategy used by companies listed on the Mexican Stock Exchange (BVM), the São Paulo Stock Exchange (VVSP), and the Santiago Stock Exchange (BCS). To this end, the data observed in the Economática database for a sample of 29 companies were considered. Then, through a long panel data model, the study concludes that in the organizations reviewed, there is a degree of association between the variables "short-term liabilities" and "share price", as the former increases by 1%, and the value of the shares increases by 0.09% in the subsequent period. This confirms a procyclical financial leverage.

**Keywords:** fluctuations and economic cycles; financial crises; financing policy and company value; illegal conduct

## 1. Introduction

Financialization has been analyzed from a wide range of approaches and disciplines (Iliopoulos and Wójcik 2021; Klinge et al. 2021; Mader et al. 2020; Reis and Oliveira 2021). For the case of company valuation in the stock market, one of the relevant empirical facts is the process of financial accumulation through the mechanism of short-term leverage (Alter and Elekdag 2020; Toporowski 2005). This is a characteristic mechanism of the financialization processes, whose primary objective is to increase the company's valuation based on the debt obtained in the short term and thus continue with the accumulation of financial capital. In the case of Latin American countries, leverage strategies have a particular characteristic: they are short term and in dollars (Rabinovich and Pérez Artica 2022; Villavicencio 2021). By leveraging in dollars, companies back these operations with speculative positions in derivatives, which deepens the process of financialization in the region.

The characterization of these strategies has some methodological problems. These problems are due to the limitations of a consistent theoretical contrast that points out the dynamics under which these strategies operate. Furthermore, and concerning the core of this problem, the processes of financial accumulation through the speculation of financial positions in dollars are an empirical fact that discards orthodox financial, economic theory. By dismissing this discussion in the orthodox financial literature, an opportunity opens to characterize the microeconomic level and make visible the financial strategies followed at the corporate level. These decisions are reflected in the company's financial position, i.e., its balance sheet and income statement. Operationally, what occurs with the accumulation of financial resources resulting from a short-term leverage strategy and in dollars in the case of some corporations in emerging economies, is tracked over time.

It is important to characterize the object of study from the methodological point of view. First, assuming that there is a cumulative process in corporate valuations, the evidence should show that this variable has a cumulative trend over time. Such a process is

unacceptable in conventional financial models, since, in such theories, cumulative processes of stock prices are considered impossible (Gromb and Vayanos 2010; Schnabel and Shin 2004). From the well-known position of Modigliani and Miller (1958), one of the factors that restrestrictricts this type of behavior in the variables is price arbitrage if stock prices follow a random trend. Thus, we are presented with a methodological problem with theoretical implications regarding the company's valuation process and the possibility of establishing a market strategy to adjust its valuation. Therefore, the usual analytical tools have restrictions, making it necessary to analyze this phenomenon with a methodology that allows controlling the endogeneity of the variable.

To evaluate the performance of speculative strategies using derivatives and their relationship with the stock market value of corporations, we have decided to use the stock market value as analyzed by Allayannis and Weston 2001; Schiozer and Saito 2009. Analyses (Cid Aranda et al. 2017; Giraldo-Prieto et al. 2017; Pinilla Bedoya and Muñoz 2020) in Latin America regularly use "Tobin's Q" to analyze the relationship between valuation and short-term leverage hedged by derivatives (Allayannis and Weston 2001). In contrast, we have discarded the analysis through the construction of Tobin's Q, since this indicator has limitations in that it assumes the liquidity of various assets is homogeneous. The analysis is based on a perspective (Toporowski 2005) where the liquidity of assets within corporations is heterogeneous ant assets function as collateral, influencing the share price and accounting for accumulation processes in financial valuation.

The choice of the period of analysis for this study is determined by the boom of the financial cycle between 2006 and 2010. The 2006–2010 period allows us to analyze the boom stage of the financial cycle closest to our object of study, i.e., financialization based on short-term dollar leverage strategies. This period is relevant because, in some Latin American countries, the speculative strategy of derivatives applied by several corporations generated financial problems, including the bankruptcy of companies. For example, during the mortgage boom and crisis in 2008, companies listed on the stock exchange of Mexico, Brazil, and Chile—Gruma, Bachoco, Cemex, Comercial Mexicana, Sadia, and Aracruz, among others—used derivatives through short-term financing to profit from financial speculation (Castro 2009; Jaramillo et al. 2012; Olivares and Jaramillo 2016).

This text begins with a section that reviews the literature on the financialization of corporations which allows the authors to raise the study's hypotheses, which will be the basis for the econometric study on financialization based on debt operations and derivatives concerning the exchange rate in the short term. The methodological section explains how the problem suggested in this section about the cumulative process of the data will be overcome. Finally, the results of the hypotheses are shown, and the study concludes with some points that clarify the process of financialization that affects the value of companies in Latin America.

## 2. Consolidation of the Financialization Process in Corporations

Corporate strategy at the global level has been determined by adopting the logic of shareholder value maximization, whose main objective is to increase financial returns (Lazonick and O'sullivan 2000; Boyer 2000). As a result, corporations designed novel ways to manage the liquidity of their portfolio to maximize short-term returns. Maximizing returns in the short term has been performed by creating financial instruments and consolidating speculative positions to obtain extraordinary returns with a narrative of risk control (Lazonick and O'sullivan 2000; Funk and Hirschman 2014; Persons and Warther 1997). Financing these positions changed the liquidity and returns in the stock market by consolidating positions in companies in derivatives to control risk, repurchase agreements to manage liquidity, and share buybacks to maximize the return on investment made by shareholders (Lazonick and O'sullivan 2000; Boyer 2000; Funk and Hirschman 2014; Orhangazi 2008; MacKenzie and Millo 2003; Andersson et al. 2010; Ezzamel et al. 2008). Short-term strategies that modify liquidity and returns have thus followed the logic of maximizing shareholder value by managing the financial position of corporations.

Changes at the corporate level have increased the demand for short-term financing and, in some cases, foreign currency positions have increased. Moreover, in companies from emerging countries, the literature shows these positions have an important component of short-term liabilities denominated in foreign currency (Alter and Elekdag 2020; Bräuning and Ivashina 2020). This implies an additional risk, considering that, in emerging countries, some corporations have limitations in immediately accessing reserve currency markets. The phenomenon is particularly important in emerging economies where capital reversals and capital flight have occurred (Schiozer and Saito 2009). This greater exposure to the risk assumed by corporations is reflected in the importance of analyzing episodes where the processes of global financial instability increase the level of exposure to bankruptcies, especially when companies are oriented towards generating speculative strategies during the boom of the financial cycle. Such behavior shows corporate social irresponsibility, as it destroys the long-term value of corporations (Willmott et al. 2016).

The hypotheses of the study are based on the literature on derivative hedging strategies. The literature has shown that this strategy is directly related to the level of valuation of the corporation. Short-term financing strategies to raise the stock price in the stock market are presented as empirically regular during the boom of the financial cycle (Toporowski 2005; Holmstrom 2015). The first study hypothesis argues that:

**H1.** *The short-term leverage of the corporations studied behaves procyclically.*

Moreover, it has been observed that, during this process, corporations orient their strategies to take advantage of the boom phase. In this sense, the literature on the financial cycle has shown that the financial positions of corporations change about the perspective of the global financial cycle (Dell'Arriccia et al. 2012; Jordá et al. 2011). Short-term financing through credit has shown an important role in the financial position of companies, which leads to procyclical behavior (Holmstrom 2015; Rungcharoenkitkul et al. 2019).

The problem is that this type of strategy, in countries with a dependence on a reserve currency, can increase the risk exposure of companies that have short-term positions in foreign currency (Han 2021). Thus, the literature has shown that in emerging markets the transmission of the global cycle operates through the exchange rate channel and on the fragility of the financial position of companies. Thus, analyses have been carried out considering financial leverage and hedging, since the latter increases the possibility of generating financial instability (Hansen and Hyde 2010).

These analyses have shown that the cycle is characterized by a stage of abundant liquidity that generates expectations in the financial system. This generates expectations beyond the limits of the rationality posed by conventional financial theory (Aikman et al. 2015; Bernanke 1993; Schularick and Taylor 2012). Expectations emerging in the financial cycle when abundant liquidity generates incentives to design corporate strategies that allow obtaining an advantage in the short term, giving greater weight to short-term financial returns (Han 2021; Beneish et al. 2013; Amiram et al. 2018; Fich and Shivdasani 2007). In the case of corporations, evidence indicates that hedging attempts to create perception based on narratives that seek to improve shareholders' financial performance in the short term (Gómez-González et al. 2012; Aretz et al. 2007; DeMarzo and Duffie 1995). In this case, this analysis will be carried out in which the narrative described above shows the greater access to short-term leverage to finance speculative positions.

In Latin America, the literature shows a relationship between direct credit and company performance during the financial cycle (Román de la Sancha et al. 2019; Berggrun et al. 2020; Krznar and Matheson 2017). In this paper, the approach links the leverage position and its valuation in the capital market. As a result, the debt capacity is related to the company's valuation in the financial markets and the search for short-term returns for institutional investors and banks during the boom of the financial cycle (Rabinovich and Pérez Artica 2022; Lazonick and O'sullivan 2000; Krznar and Matheson 2017; Dore 2008; Rey 2015). Under the financialization approach, the financial cycle deepens, the boom is prolonged, and the bust is deeper. This can explain why credit is procyclical and

determined by the company's need for liquidity. The growing interest in the literature on dollar leverage levels in emerging markets motivates the analysis, especially in Mexico, due to its level of exposure in dollars.

The second hypothesis advance a step forward trying to link the short-term pro-cycle companies' strategies with solvency and financial fragility and instability problems, that is:

**H2.** *Latin American corporations' short-term derivative trading strategies generate financial fragility.*

It has been identified that certain speculative strategies to increase financial performance are systematically used by companies, for example, in operations with financial derivatives, which are associated with high risk. When the financial cycle ends, companies are forced to sell their domestic currency positions, weakening the local currency and putting pressure on the exchange rate, and as a result, the solvency of corporations deteriorates. This phenomenon can generate a vicious circle, where foreign currency liabilities increase, and greater losses accumulate.

In the case of the derivative markets in Brazil and Chile, there is a robust over-the-counter market. According to Dodd and Griffith-Jones (2006) the operations of these markets reach one billion dollars daily. In the case of Chile, the main negotiators are Santander, JP Morgan, and Deutsche Bank. Most of these operations are based on a dynamic hedging strategy, which generates a procyclical behavior, where instability appears in the exchange markets, and pressures are generated from contracts and hedges. In Chile, the literature points out that these strategies generated instability in the financial markets in 2008, caused by the short-term financial fragility to which companies were exposed (Schiozer and Saito 2009; Cid Aranda et al. 2017; Dodd and Griffith-Jones 2006).

In the literature, the discussion is focused on the choice made by companies on how they use derivatives for hedging or speculation (Allayannis and Weston 2001; Dodd and Griffith-Jones 2006). The problem is that these strategies generate "financial fragility" since the exposure to liquidity problems increases within companies, and the possibility of defaulting on payment commitments in the short- and long-term increases (Tsomocos 2003). The origin of the literature on financial fragility begins with the problems in Latin America regarding exchange rate volatility (Eichengreen and Hausmann 1999; Foley 2003). Based on this literature and the tequila effect crisis in Mexico in 1995 and its contagion to emerging countries, a series of innovations were developed in the financial sector in which the use of short-term derivative transactions was considered. Such use is focused on the speculative use of derivatives (Farhi and Borghi 2009). It is in Mexico and Brazil where the issue of financial fragility reappeared when two companies went bankrupt due to the operation of exchange rate derivatives. Thus, corporations negatively affect stakeholders in the company—such as workers and suppliers, to name a few—for the gain of equity stakeholders. To detect this strategy, it is important to consider that companies use derivatives in this way; the literature in Latin America shows that this speculation strategy was used in Brazil, Mexico, and Chile (Dodd and Griffith-Jones 2006; Farhi 2017; Farhi and Borghi 2009; De Souza Murcia et al. 2017). This can be seen through an increase in the foreign currency liabilities associated with the position, which generates fragility in the face of exchange rate movements.

The final inquiry of this study raises the hypothesis of why Latin American companies increase their foreign currency liabilities, accepting fragility in the face of exchange rate movements. That is:

**H3.** *Latin American corporations establish strategies through speculative operations that raise the company's financial valuation.*

Expenses are significant, with (Kamil 2008) showing that 2003 transactions represented 53.2% of the GDP in Mexico and 225% of the GDP in Chile. In the case of Chile, literature has identified that exporting companies use exchange rate hedges to cover their positions (Chan-Lau 2005). In fact, due to the pressures on the global financial cycle and the sovereign

debt problems in Argentina during 2001, increased volatility in Chile was attributed to multinational firms hedging short-term dollar funding positions.

An analysis of 720 large companies in the USA between 1990 and 1995 shows that companies that use derivatives have a higher market value and report an increase of 4.87% in their market value (Allayannis and Weston 2001). However, it is important to ask about the causality of this relationship. Particularly when studying larger companies, it is not clear whether these corporations engage in derivative transactions because they are outperforming companies or vice versa. On the other hand, regarding the relationship between derivatives operations and debt capacity, (Graham and Rogers 2002) found that they increase profits and the demand for higher financing. Regarding derivatives trading and its positive impact on market valuation, Corona Dueñas (2012) finds a positive relationship considering the BMV price quotation index.

The choice of the methodological strategy and the characterization of some Latin American corporations aims to evaluate the financialization process using debt and foreign currency derivatives in the short term. Companies moved from a hedging position to a speculative place to raise the company using debt and foreign currency derivatives in the short term. Companies moved from a hedging position to a theoretical place to grow the company's valuation and returns. Operations with various hedging instruments that were used speculatively have allowed for the obtaining of returns that are not the result of the company's operating activities (Farhi 2017; Farhi and Borghi 2009).

The analysis will be carried out by estimating an econometric panel model for a sample of companies in Mexico, Brazil, and Chile during the boom period of the financial cycle between 2005 and 2008. In this way, the study will generate evidence on structural practices within the context of a sample of companies in Latin America to understand the impact of short-term financing strategies and the use of hedging through derivative instruments as a strategy to increase the value of the company in the stock market.

### 3. Materials, Methods, and Model Specification

The construction of the stylized model was intended to identify corporations that borrowed in dollars to finance speculative positions in derivatives during the boom and financial fragility, before positions in derivatives during the crash and financial fragility before the great financial crisis of 2008. In the literature review, we focused on Latin America, and specifically on three countries: Brazil, Chile, and Mexico. This criterion is because the literature has pointed out that short-term speculative strategies financed with dollars in these countries were widespread in companies with derivative positions. This way, a search was carried out, the main criteria of which were corporations with short-term financing records in dollars and a place in derivatives. The study data were obtained from the Economática database. The selection criteria yielded 29 corporations (Appendix A) between 2006 and 2010, with a quarterly periodicity, that is, 20 quarters representing a total combination of 551 observations for the construction of the stylized facts and the estimation of a long panel.

The variables used for analysis were:

($LPEXT$) Logarithm of short-term liabilities in foreign currency.

($LPA$) Logarithm of the share price.

($LVIX$) Logarithm of VIX.

($LEMBI$) Logarithm of the country risk for Chile, Brazil, and Mexico.

($LRBG$) Logarithm of the company's operating results

Empirical evidence using an econometric panel model

To generalize the analysis, we proceeded to use a formal test of the relationship between short-term liabilities and the market value of companies with the following model:

$$LPA = C + \beta_1 LPEXT_{it} + \beta_2 LPA_{it-k} + \beta_3 LVIX_{it} + U_{it}$$

where $C$ is the constant, and $\beta$ are coefficients associated with each variable, the subscript $t$ = the periods, and $i$ = the cross sections. Under the assumptions raised, a significant negative effect of $(\beta_1, \beta_2)$ and a negative and significant effect of $(\beta_3)$ were expected.

The modeling process required controlling for endogeneity issues; furthermore, due to the assumptions made, the objective was to determine if there was a cumulative process of the dependent variable, i.e., if an increase in LPA in the previous quarter had a concurrent effect on LPA. It has recently been shown that controlling for endogeneity is central to the presentation of results, especially when specification errors such as omitted variables are considered (De Souza Murcia et al. 2017). The endogeneity problem is central to econometric modeling and must be controlled with theoretical and exploratory validation of the data (Ullah et al. 2020). Detecting the endogeneity trap requires a series of steps, the first of which is to detect, with the support of the theory, the relevant variables to explain the error perturbations. The problem is demonstrating that the variables identified as exogenous are exogenous and that these variables are not related to the errors. Similarly, controlling for endogeneity is useful when a variable whose value accumulates over time is included, as suggested by the work of (Ketokivi and McIntosh 2017).

Although instrumental variables solve certain problems concerning endogeneity control, they generate problems concerning the loss of efficiency of the estimators (Ullah et al. 2020). Therefore, modeling endogeneity in the literature has become a central issue for this type of methodology since the estimators are unstable, and the results depend on the characteristics of the sample (De Souza Murcia et al. 2017; Ullah et al. 2020; Lu et al. 2018). In addition, the incorporation of lagged variables generates problems of serial correlation in the errors, such as those resulting from the estimation of the model and the errors of the fixed effects related to the dependent variable with a lag.

However, endogeneity can be controlled by using instrumental variables, which depend on the estimator and the specification used to assess whether the lagged variable is related to the contemporaneous variable (De Souza Murcia et al. 2017). The literature points out that the two-step estimation is more efficient by using the weighting of a heteroscedastic matrix for the periods of the generalized method of moments (GMM) developed by Arellano and Bond (Ullah et al. 2020; Bun and Sarafidis 2015). The choice of model depends on the over-identification of the instrumental variables used. In this way, instrumental variables are incorporated since they may be correlated with other regressors in the model and may explain the behavior of the stock's valuation. Applying the methodology of (Ullah et al. 2020) the theoretical arguments for the selection of instrumental variables were considered.

Following the Ketokivi–McIntosh procedure, the lagged effect of LPA shares and LPEXT foreign currency liabilities were used as instrumental variables. These variables capture the cumulative behavior of the financing and valuation process, as pointed out in the work of (Borio 2014). In the case of the other variables of the model, the financial cycle represented by the EMBI (Emerging Markets Bonds Index) was introduced for each country; the choice was the result of the review of the literature on the financial cycle, in which the EMBI is an important component to explain how volatility is transmitted globally (Foley 2003; Borio 2014).

The estimation of an extended panel data model required verifying whether the overidentification of the instrumental variables was valid (Ullah et al. 2020; Labra and Torrecillas 2018) by performing the following hypothesis test:

Ho: the restrictions of over identification are valid.

Ha: the restrictions of over-identification are not valid.

In the specified model, the range of instruments was (29) and the coefficients were (3); therefore, there was overidentification of constraints in the estimated model (29 > 3). The Sargan test was constructed to verify this overidentification. The test was distributed as a chi-square, which was distributed as follows: $\chi^2 \sim (j, p - k)$. In the results of the dynamic panel model, the statistic $j$ was obtained from the Sargan estimation, $(p)$ was the number of instruments, and $(k)$ the number of coefficients. In the estimation performed,

the probability of committing the type I error was very high (0.8984); therefore, Ho was accepted, which shows that the overidentification restrictions were valid (see Appendix B).

As mentioned above, the second dynamic panel model specification problem was the serial correlation of the errors (Labra and Torrecillas 2018). To eliminate autocorrelation within the cross sections and increase the robustness of the results, White's period weighting was used. In this model the criteria to be tested was:

Ho: There is no correlation.

Ha: There is correlation.

The Arellano–Bond test showed that the error terms were not correlated in levels, so we cannot reject the null hypothesis in AR (2) with a probability of 0.8202 (See Appendix B). The rejection criterion indicates that in the AR (2) process, the probability of committing the type I error is high, and the null hypothesis must be accepted; therefore, the errors are not correlated. As a result, given that the overidentification restrictions are valid and the errors are not correlated, it was established that the use of instrumental variables to control endogeneity is correct and allows for obtaining robust results for the analysis performed.

## 4. Results

The structure of the panel information allows us to graph the average behavior of the 29 companies during the period (see Figure 1).

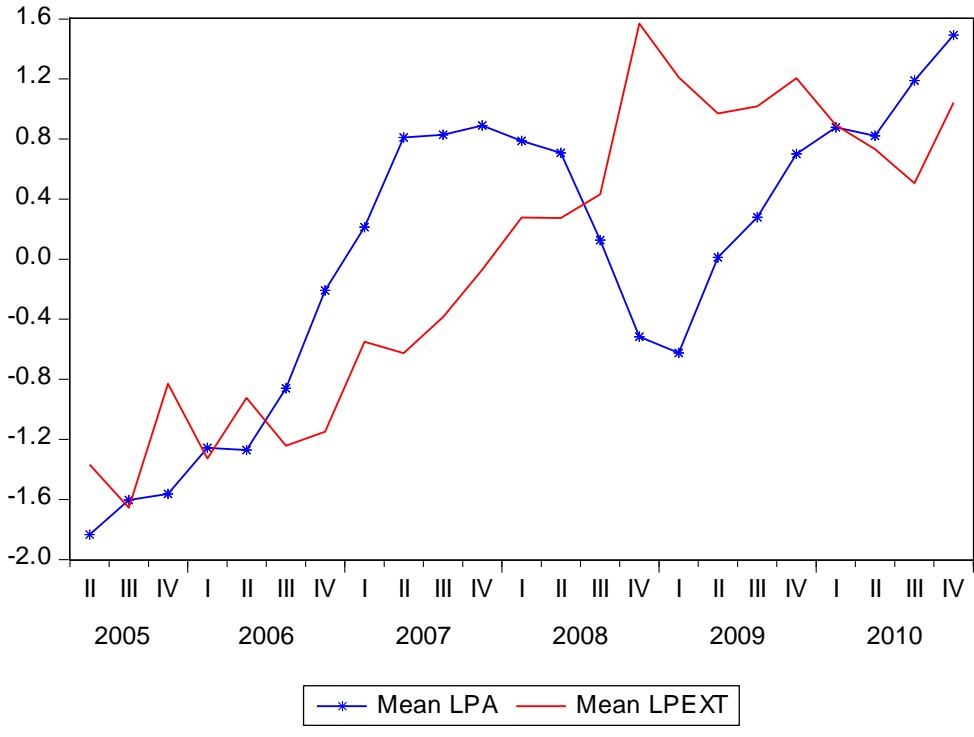

**Figure 1.** Average behavior of the 29 companies during the period.

As of 2005, the data show an increasing trend in short-term liabilities in foreign currency, a trend that accelerated in the last quarter of 2008. During this period, Latin American companies carried out derivative transactions mainly with foreign banks such as JP Morgan, Lehman Brothers, and Credit Suisse (De Souza Murcia et al. 2017).

The accelerated increase in liabilities at the beginning of 2008, in nominal terms, is due to the decrease in the exchange rate, which in fact coincides with the fall in the share price. Regarding the relationship between share price and short-term liabilities, an increasing trend is observed on average for the 29 companies, which accelerates as of 2005 along with the increase in liabilities; in fact, the increasing trend in share price continues until the second quarter of 2008. The companies with the greatest exposure to losses were Cemex with US$700 million, Gruma with US$684 million, and Comercial Mexicana, Sadia, and

Aracruz, each with a loss of US$1 billion. These last three corporations filed for bankruptcy, an event that justifies the selection of the period, since bankruptcies due to exposure to financial risks are unlikely for large corporations in Latin America (See Figure 2).

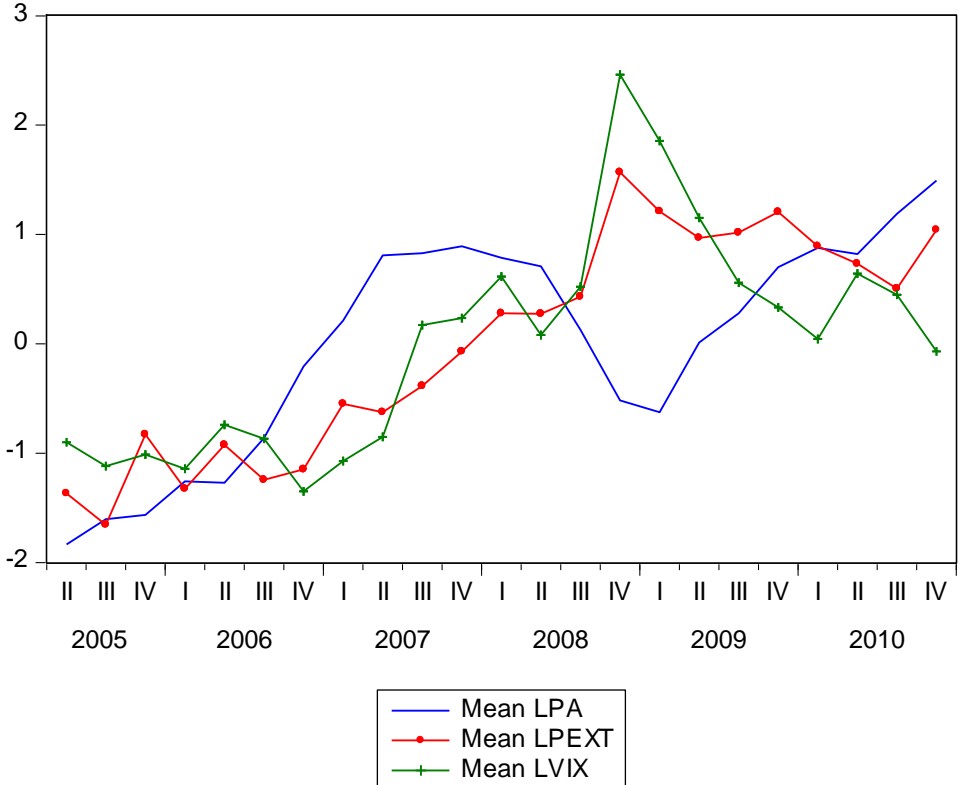

**Figure 2.** Average behavior of the 29 companies during the period.

For example, in the specific case of the Sadia corporation, the chronicles report that the company justified its derivative positions, stating that they were a hedge for its income in dollars from foreign sales. One of the problems is that the corporation's bylaws stated that exposure to dollar-denominated debt in the short term was capped at US$1.8 billion, a figure that represented 6 months of accumulated export revenue (De Souza Murcia et al. 2017). Once the exchange rate depreciated and the derivatives strategy failed, the company was exposed for US$7.6 billion; in fact, total debt went from representing 3.7 times to 6.7 times EBITDA during the increase in the value of foreign currency liabilities (De Souza Murcia et al. 2017). The regulatory investigations concluded that the corporation had no way of recording such exposure to financial fragility. According to the information, the company failed to comply with regulations by recording derivative transactions as hedges, since they were used for speculative purposes. As pointed out by some media, the company became a financial institution that established strategies to increase the company's valuation and returns and profits for shareholders (De Souza Murcia et al. 2017).

The estimation results show a positive and significant relationship between short-term liabilities (LPAST) and stock prices (LPA). When liabilities increase by 1% in the previous period, the value of shares increases by 0.09%. Thus, the first hypothesis, where leverage has a procyclical tendency, is proved (See Table 1).

**Table 1.** Econometric model results.

| Var. Explained | LPA$_t$ |
|---|---|
| | GMM Method |
| Explanatory | 2 steps |
| LPA$_{t-1}$ | 0.8382 *** |
| | (0.0066) |
| PASEXT$_t$ | 0.0905 ** |
| | (0.0069) |
| LVIX$_t$ | −0.1528 |
| | (0.0164) |
| Observations | 551 |
| J-statistic | 28.25 |

*** $p < 0.01$, ** $p < 0.05$.

The model data show that during the boom of the financial cycle between 2005 and 2007, the corporations analyzed, on average, accumulated liabilities in foreign currency at a growth rate of 128%, i.e., it would have a positive effect on the stock equivalent to a growth of 11.52%, according to the results of the model estimation. The 11.52% increase in shares today would mean an equivalent increase in the following period of 9.5% due to the inertial effect of the variable. In the case of the VIX at the end of the boom, losses of 29% were accumulated with a negative effect on the stock of 4.35%.

Regarding the endogeneity of the independent variable (LPA) lagged one period, the results show that a contemporaneous 1% increase in the share price will have a positive impact of 0.83% in the following period, i.e., the share price has an inertial behavior. As the company establishes strategies with derivative instruments, the valuation of shares increases, and a period later has a cumulative effect on the same valuation.

Finally, the LVIX variable has a significant but negative impact, as a 1% increase in overall financial volatility reduces the company's valuation by 0.15%. This result shows the impact of the global financial cycle when instability processes rise. In addition, this result shows that the methodology for choosing instrumental variables based on theory improves the estimation process of the 2-stage dynamic panel econometric models.

A relevant limitation of this study is the use of data from a limited period, from 2006 to 2010. However, the selection of these years is based on the opportunity that the period offers for studying the financialization processes of Mexican companies during periods of economic boom. Therefore, the results are relevant to the hypotheses and the question that motivated the research. Moreover, a second limitation of the study is that we have less evidence on the use of derivatives and their impact on corporate valuation in Latin America because a limited number of companies export goods and take positions in those subordinate financial products. Furthermore, in some cases, such as Brazil, there is a regulation on debt positions in dollars (Gómez-González et al. 2012; Schiozer and Saito 2009).

## 5. Discussion and Conclusions

The estimation results corroborate the first hypothesis regarding the effect of leverage on the valuation of corporations in the short term. As the literature on financialization has shown in the works of (Toporowski 2005; Holmstrom 2015; Rungcharoenkitkul et al. 2019) the procyclicality of credit is an empirical fact, and, in this case, it has been proven in the behavior of a sample of corporations. Hypothesis 1, which establishes that the short-term leverage of the corporations studied has a procyclical behavior, is thus verified. This result aligns with the literature showing a direct relationship between credit and firm performance over the financial cycle (Román de la Sancha et al. 2019; Krznar and Matheson 2017). Companies operate derivatives in a procyclical manner, in which transnationals increase their short-term indebtedness capacity, including in foreign currency, as they become more profitable. This entails operating through hedging, which transitions from a risk hedge to a speculative hedge. Seen as a phenomenon of financialization, the fact that

credit and corporate performance is procyclical shows that the financial cycle deepens, the boom is prolonged, and the bust sharpens.

Results contradict conventional financial theory since credit increases despite the increase in risk exposure. In fact, conventional theories of borrowing capacity have serious limitations in explaining the propensity of lenders to finance risky positions to obtain higher returns on their investment or credit placement. The classic positions on risk present important restrictions to the explanation of the phenomenon of irresponsibility incurred by companies to increase shareholder value to the detriment of other stakeholders. Higher return on stock generates expectations that allow a high-risk position to be taken, financed based on speculative instruments that enable gaining some advantage but exacerbate the boom and fall of the financial cycle.

The study presents new evidence that corroborates that foreign currency financing and its hedging through exchange rate derivatives have a positive impact on company valuation. This result is in line with the work of several authors (Corona Dueñas 2012; Gómez-González et al. 2012; Han 2021; Aretz et al. 2007; Bun and Sarafidis 2015; Geyer-Klingeberg et al. 2021). This type of strategy is used to increase the valuation of its shares in the stock market and was used by the corporations analyzed, which allows us to obtain important information on the strategy used. This choice, on the part of the companies, shows that companies increase their level of exposure and thus face greater liquidity problems, which increase the possibility of defaulting on payment commitments in the short and long term (Tsomocos 2003). Hypothesis 2 is thus corroborated since these strategies, carried out through short-term derivative transactions, generate financial fragility in Latin American corporations.

As mentioned in the literature, companies that use derivatives in the United States report an increase of 4.87% in their market value (Allayannis and Weston 2001). The results of the present study corroborate this relationship since the share price shows inertial behavior, i.e., as the company establishes strategies with derivative instruments, the valuation of the shares increases, and a period later has a cumulative effect on the same valuation. This result supports the characterization of financial inflation and cumulative capital processes in stock markets (Toporowski 2005). When instability processes are elevated during the cycle, it harms the overall financial cycle during the downward phase of the financial cycle boom, in agreement with the works of (Alter and Elekdag 2020; Tulum and Lazonick 2018; Bräuning and Ivashina 2020; Han 2021; Krznar and Matheson 2017; Rey 2015; Yan and Huang 2020). Regarding its use as an instrumental variable, it shows that the variables selected from a theoretical perspective can improve the estimation processes of the 2-stage dynamic panel econometric models.

The overall results for a selection of Mexican companies show that corporations establish speculative positions based on their short-term foreign currency financing strategies. The strategy aims to accumulate short-term liabilities in a procyclical manner in the boom of the financial cycle; this is a behavior already pointed out in the literature (Toporowski 2005). The analysis provides empirical evidence on how rising stock returns generate expectations that allow taking a risky position financed by a speculative strategy in the corporation, which exacerbates the boom and bust of the financial cycle. This aggressive search for higher shareholder returns (Lazonick and O'sullivan 2000; Whitley 2003; Archel Domench and Villegas 2014) is widely observed in the selection of companies in Latin America, with the potential for corporate governance to be subordinated to the stock market gains posited by the financialization theory. At the same time, the empirical data studied show that in the countries of the region, corporations follow a strategy linked to the establishment of fragile financial positions that deteriorate rapidly when the local currency is devalued, with immediate effects on the value of the share (Baines and Hager 2021; Schwartz 2021). Similarly, as (Hansen and Hyde 2010) showed in the case of LA, the exposure to risk entails a possible bankruptcy in the face of volatility processes transmitted through the exchange rate in emerging economies.

As a corollary, the results support Hypotheses 2 and 3 and call for active policy regulation of these short-term operations. At the same time, the results call for a corporative government that demands an effective approach to financial company policies. Policy solutions demand a balanced approach between government and governance policies. Corporations need internal strategy governance regulations of their speculative operations that look to raise the company's financial valuation at the expense of their financial fragility; this is a problem that also demands active authority policy regulations.

In parallel, this study also contributes to the methodological side. In this vein, the article's identification of the impact of a financial variable with a cumulative process to explain the financialization process is a contribution. Furthermore, the use of instrumental variables to control the endogeneity of the variables in the estimation of the model, where theoretical criteria were used following the contributions of (Ullah et al. 2020), made it possible to identify the impact of the financial cycle on the speculative financial position applied by various corporations in Latin America, in the context of global financialization.

All in all, the results and the approach of this article have some limitations. In particular, and concerning the study period, it is necessary to evaluate whether, during a period of normality, the variables considered have the same impact on the value of the share in the stock market. From the same perspective, the selection of corporations is limited to those that report a liability position in dollars and that use derivatives; thus, the indicated behavior is generalized in this selection of corporations. Moreover, as this study explained before, it is necessary to emphasize that the analysis is valid during the boom period when there is an accumulation of foreign currency liabilities and hedging and speculation strategies concerning this financial position. In the case of Latin America, it is difficult to generalize the results to most publicly traded corporations since there are corporations with conservative strategies that are not exposed to global risks.

As the selection presents limitations based on the obtention of financial information, and detailed information can only be obtained in cases where companies have been accused of fraud, there are possible extensions of this work considering that corporate strategies oriented exclusively to financial profitability are unsustainable in the long term. Studies that advance such an endeavor will lead to a better understanding of Latin American financialization processes and to potential improvement in financial market policy and corporate governance recommendations.

**Author Contributions:** Conceptualization, J.F. and J.F.E.-C.; methodology, J.F. and N.L.; software, J.F. and L.L.; validation, J.F., J.F.E.-C. and N.L.; formal analysis, J.F.; investigation, J.F.E.-C.; resources, J.F. and N.L.; data curation, J.F. and L.L.; writing—original draft preparation, J.F.; writing—review and editing, J.F.E.-C.; visualization, J.F. and J.F.E.-C.; supervision, J.F.E.-C.; project administration, J.F.; funding acquisition, J.F.E.-C. All authors have read and agreed to the published version of the manuscript.

**Funding:** This article was possible by the support of Proyecto DGAPA UNAM PAPIIT IA303919: "Ciclo financiero. Enfoques, retos y perspectivas en la gestión financiera de las corporaciones y el diseño de la política macroeconómica".

**Informed Consent Statement:** Not applicable.

**Data Availability Statement:** Data for the present paper is available in this link: https://shorturl.at/ beik2 accessed on 1 September 2022.

**Conflicts of Interest:** The authors declare no conflict of interest.

## Appendix A

| Mexican Corporations | Brazilian Corporations | Chilean Corporations |
|---|---|---|
| GGIGANTE | EMBRAER | ANTARES |
| GMEXICO | SADIA | SCHWAGER |
| MOCTEZUMA | ARACRUZ | LATAM |
| ELEKTRA | LOJA | FALABELLA |
| TELEVISA | KLABIN | CENCOSUD |
| G TMM | SAO MARTINO | ALMENDRAL |
| LIVEPOOL | ALPARGATAS | AES GENER |
| MEXCHEM | CESP | BLUMAR |
| PEÑOLES | | |
| POSADAS | | |
| HERDEZ | | |

## Appendix B

Sargan Test
Instrument specification:
@DYN(LPA,-2) LEXCP LOGRBG LTDC LIPC

| J-statistic | Instrument rank | Prob(Jstatistic) |
|---|---|---|
| 28.25720 | 29 | 0.8984 |

Arellano–Bond Serial Correlation Test
Included observations: 251

| Test order | m-Statistic | rho | SE(rho) | Prob. |
|---|---|---|---|---|
| AR(1) | −3.776010 | −13.695326 | 3.626931 | 0.0002 |
| AR(2) | −0.227225 | −0.509775 | 2.243480 | 0.8202 |

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
