# Peer review of "Inappropriate Corporate Strategies: Latin American Companies That Increase Their Value by Short-Term Liabilities"

_ijfs, doi:10.3390/ijfs10040100_

Round 1
Reviewer 1 Report
The work is interesting. Various features of the work are appreciable.
The most critical points are in my opinion:
- too small sample (the number of companies needs to be increased)
- not particularly long observation time
Suggestions:
- to remedy the two main critical points
also
- to place the research hypotheses in the literature paragraph, with each hypothesis being followed by the specific literature
- discussions and conclusions can be placed in the same paragraph, perhaps implementing policy implications
Author Response
Manuscript ID 1930383
Juan Felipe Espinosa-Cristia
Departamento de Ingeniería Comercial, Universidad Santa María, Chile;
Valparaíso, Chile
10-10-22
Dear Reviewer,
We appreciate the opportunity to amend our article based on your suggestions and comments.
Below we offer our responses and detail about the changes in our article.
Reviewer 1
|
N° |
Commentary |
Responses and Changes in the Text |
|
1 |
Is the research design appropriate? Can be improved |
Thanks for the recommendation. We worked to clarify the research design explaining the limitations and assumptions of the study. |
|
2 |
Are the methods adequately described? Can be improved |
Thanks for the recommendation. We worked to clarify the methods section explaining the limitations and assumptions of the study. |
|
3 |
Are the conclusions supported by the results? Can be improved |
Thanks for the recommendation. We worked on generating a whole section connecting discussion and conclusions in one place. |
|
s4 |
- too small a sample (the number of companies needs to be increased) |
Thanks for the comment. The object of study of the article is based on companies in Latin America that will be listed on the stock market and that have relevant data for the required variables (they have liabilities in foreign currency and derivatives). Among the companies that meet these characteristics, we have the 29 companies compiled in the Economática database. This way, the sample size coincides with the entire universe of organizations investigated in the three stock exchanges.
|
|
5 |
- not particularly long observation time |
Thanks for the comment. From our technical perspective, we consider that the panel is a long one in terms of time, and the specification was correctly used with n>100 and t>15 Literature express that when ‘t’ is greater than 15, it is already a long panel.
|
|
6 |
(…) place the research hypotheses in the literature paragraph, with each hypothesis being followed by the specific literature |
Many thanks for the comment. We changed the order and organized the text emphasizing the literature that supports all our article’s hypotheses. |
|
7 |
discussions and conclusions can be placed in the same paragraph, perhaps implementing policy implications |
Thanks for this critical point. We changed and reorganized the discussion and conclusion section, and, at the same time, we elaborated on some policy implications. |
All the best,
Dr. Juan Felipe Espinosa Cristia
In the name of the authors’ team
Reviewer 2 Report
This paper manages to make a considerable contribution to the investigation of the variables “short term liabilities” and “share price” in 29 Latin American countries. The topic considers the economic cyclicity and operates with data from the mortgage boom and crisis in 2008.
The paper is well structured and innovative. Literature review is actual and is considering some historical experiences in order to better sustain the arguments of the research and conclusions drawn at the end of the paper. The authors demonstrate good knowledge of the most important papers written on the theory of stock markets, fluctuations and financial crisis, financial policies and their impact on corporations. The research question is well highlighted with clear hypothesis reflected in the paper structure.
The aim of the paper and the methodology are clearly defined. Both quantitative and qualitative methods are explained in the paper. Using the econometric model, it is demonstrated the positive correlation between “short term liabilities” and “share price”.
Interpretations and conclusions are justified by the results and are useful for both theory and practice. Also, the limits of the analysis are well presented and the proposals for future directions of the research are suitable.
Author Response
Manuscript ID 1930383
Juan Felipe Espinosa-Cristia
Departamento de Ingeniería Comercial, Universidad Santa María, Chile;
Valparaíso, Chile
10-10-22
Dear Reviewer,
We appreciate the opportunity to amend our article based on your suggestions and comments.
Below we offer our responses and detail about the changes in our article.
|
N° |
Commentary |
Responses and Changes in the Text |
|
|
English language and style are fine/minor spell check required |
We checked again English spelling, usage, and style. |
Finally, we want to thank your positive review of our paper.
All the best,
Dr. Juan Felipe Espinosa Cristia
In the name of the authors’ team
Round 2
Reviewer 1 Report
Thank you for your work. There remain a few perplexities about the work that I feel cannot be resolved (sample size, length of the observation period, etc.). After the revision in any case the work seems to me to have improved.